# Home-Based Stair Climbing as an Intervention for Disease Risk in Adult Females; A Controlled Study

**DOI:** 10.3390/ijerph18020603

**Published:** 2021-01-12

**Authors:** Elpida Michael, Michael J. White, Frank F. Eves

**Affiliations:** 1School of Sport, Exercise and Rehabilitation Sciences, University of Birmingham, Birmingham B15 2TT, UK; elpidamelissa@gmail.com (E.M.); M.J.White@bham.ac.uk (M.J.W.); 2School of Humanities and Social Science, University of Nicosia, Nicosia 2417, Cyprus

**Keywords:** cardiovascular disease risk, the metabolic syndrome, stair climbing, cardiorespiratory fitness, serum lipids, body composition, blood glucose, home-based vs. gym-based exercise

## Abstract

Cardiovascular disease and the metabolic syndrome are major contributors to health care expenditure. Increased physical activity reduces disease risk. The study compared effects of walking up and down stairs at home with continuous, gym-based stair climbing on the disease risk factors of aerobic fitness, serum lipids, body composition, fasting blood glucose, and resting cardiovascular variables. Sedentary women (31.7 ± 1.4 years) were randomly assigned to home-based (*n* = 26) or gym-based (*n* = 24) climbing for five days.week^−1^ over an eight-week period. Each ascent required a 32.8-m climb, with home-based climbing matching the vertical displacement in the gym. Participants progressed from two ascents.day^−1^ to five ascents.day^−1^ in weeks 7 and 8. Relative to controls, stair climbing improved aerobic fitness ($\dot{V}$O_2_max +1.63 mL.min^−1^.kg^−1^, 95% CI = 1.21–2.05), body composition (weight −0.99 kg, 95% CI = 1.38–0.60), and serum lipids (LDL cholesterol −0.20 mmol.L^−1^, 95% CI = 0.09–0.31; triglycerides −0.21 mmol.L^−1^, 95% CI = 0.15–0.27), with similar risk reductions for home and gym-based groups. Only the home-based protocol reduced fasting blood glucose. Discussion focuses on stair climbing bouts as time-efficient exercise and the potential benefits of a home-based intervention. Stair use at home offers a low-cost intervention for disease risk reduction to public health.

## 1. Introduction

### 1.1. Background

Cardiovascular disease (CVD) is a major cause of death worldwide [1]. For the UK, costs for treating CVD in England were estimated at 7% of the total National Health Service (NHS) budget, with a similar percentage of total health care expenditure for the European Union (EU), 8% [2]. For the US, CVD required 17% of all US health care expenditure [3]. For diabetes, a major component of the broader risk conveyed by the Metabolic Syndrome (MetS) [4], health care costs were estimated at 10% in the UK [5] and EU [6], with a US estimate of 14% [7]. To ameliorate these costs, risk factors can be modified by lifestyle changes. Levels of physical activity, and the associated beneficial effects on cardiorespiratory fitness, blood pressure, serum lipids and blood sugar concentrations, can be protective. Nonetheless in 2007, 94% of men and 96% of women in the UK were less active than the recommended amount [8], with similar levels of insufficient activity in the US, 89% of men and 91% of women [9]. Increased physical activity is a persistent target to improve population health in the developed world. If possible, vigorous activity will confer more protection than activity of moderate intensity [10,11,12].

Stair climbing is a vigorous activity of daily living. At 9.6 metabolic equivalents (METs) of sitting, stair climbing requires more energy per minute than jogging [13]. The seminal Harvard Alumni studies indicated that stair climbing was protective for coronary heart disease, stroke and all-cause mortality [14,15,16]. Subsequent experimental studies confirmed effects on CVD risk factors.

### 1.2. Stair Climbing Interventions and Cardiorespiratory Fitness

For estimates of cardiorespiratory fitness, all studies report improvements [17,18,19,20,21,22,23]. These consistent effects on a major risk factor for CVD are particularly encouraging; cardiorespiratory fitness has protective effects for premature morbidity and mortality [24,25,26,27,28,29]. The magnitude of risk on mortality from low fitness [Relative Risk (RR) = 1.52] was estimated as greater than risks from high blood pressure (RR = 1.30), high cholesterol (RR = 1.34), high blood sugars (RR = 1.24) and being overweight (RR = 1.02) [26]. Meta-analysis confirms protective effects of cardiorespiratory fitness on mortality and CVD in longitudinal studies [28]. Cardiorespiratory fitness is a potential mediator of the protective effect of physical activity on mortality [27,29].

### 1.3. Stair Climbing Interventions and Risk Factors for CVD and MetS

Less consistent, but encouraging, evidence of improvements in other disease risk factors with increased stair climbing has been reported. For serum lipids the pattern is mixed. While the initial controlled study reported improvements in HDL cholesterol but not LDL [18], subsequent studies reported improvements in LDL but not HDL [19] and no changes in either lipid [21]. One uncontrolled study reported improvements in LDL but not HDL [22] whereas another reported no changes [30]. No stair climbing intervention has reduced triglycerides [19,22,23,30], a major risk factor, along with low HDL and elevated fasting glucose, for the broader health risk conveyed by MetS [4,10,11]. Despite the mixed evidence, improvements in serum lipids seem a likely outcome of increased stair climbing that was tested here. For body composition, there were no changes in controlled studies whether indexed by body mass and fat from skinfolds [18], BMI [19], body mass and fat from bioelectrical impedance [21], or body mass and fat from displacement plethysmography in two studies [17]. Instead, Allison and colleagues reported an increase in fat-free mass in the second study [17], a finding that suggests possible increased ability to climb, tested here as leg power. While improvements in body composition in an uncontrolled study offered weak encouragement [22], body mass and fat, as well as blood pressure and glucose, were assessed here because of their importance to disease risk, despite insufficient evidence.

### 1.4. The Current Study

This paper reports a test of the effects of increased stair climbing at home as a potential low-cost, public health intervention. While many guidelines for physical activity suggested accumulation of 10-min bouts of at least moderate intensity will achieve cardiorespiratory benefits e.g., [31], the most recent review concluded that bouts of less than 10 min were associated with health outcomes [32]. Improvements in cardiorespiratory fitness from stair climbing can be achieved with relatively low temporal costs [17,18,19,20,21,22,23]. This study tested the effects of an eight-week intervention for increased stair climbing on aerobic fitness, leg power, serum lipids, body composition and fasting blood glucose, as well as resting cardiovascular variables, in sedentary females. The UK studies that changed serum lipids tested only females and we replicated this strategy to allow comparison [18,19]. In a major departure from previous studies, stair climbing at home was compared with an equivalent volume of stair climbing in the gym. Participants at home walked up and down the staircase in their house, choosing the speed that suited them. As in previous UK studies, a progressive increase in the number of daily climbs occurred over the intervention period [18,19,21]. Unlike most of the previous research, participants were unsupervised after the initial instruction, (c.f. [17,18,19,21,23]), a modification that reduced potential economic cost of the intervention. Home-based climbing is low-cost for participants and needs to be so for health services. As they were walking up and down a single staircase, a gym-based group compared the effects of alternating ascent and descent with the continuous climbs employed in much of previous research [18,19,21]. We predicted stair climbing would reduce all measured variables with three exceptions; increases in estimated $\dot{V}$O_2_max, leg power and HDL cholesterol were expected.

This study tested the effects of walking up and down stairs at home on a range of CVD and MetS risk factors as a low-cost intervention for public health. Increased stair climbing reduced CVD risk (LDL and HDL cholesterol, weight, aerobic fitness) and MetS risk (body fat, triglycerides, HDL cholesterol).

## 2. Methods

### 2.1. Participants

Power calculations were based on the non-significant reductions in triglycerides with stair climbing (−0.15 mmol.L^−1^; 19) as no previous study had demonstrated effects on triglycerides. With α at 0.05 and 80% power, a sample size of 43 in the experimental group would be required to demonstrate changes overall pre vs. post (GPower 3.1.9) (Kaul, Universität Kiel, Germany). As the estimate of triglyceride change was based on only eight individuals [19], inadequate to accurately test distributional properties, we erred on the side of caution and recruited 52 individuals. The study was performed in accordance with the ethical standards laid down in the 1964 Declaration of Helsinki and approval obtained from the University of Nicosia ethics subcommittee. All participants gave written informed consent.

Female volunteers (18–45 years) were recruited by email from four different companies for a study of the effects of stair climbing on physiological variables (Cyprus, 2015–2016). The International Physical Activity Questionnaire short form (IPAQ) was used to identify sedentary individuals, with a cut-point at 40 min of moderate to vigorous physical activity.wk^−1^ (MVPA) and not starting a new exercise programme in the next two months. Exclusion criteria were a history of diabetes, osteoarthritis, cardiovascular disease or other medical conditions that would impede regular stair climbing, and contraindications on the Physical Activity Readiness Questionnaire [33]. Initially, 52 volunteers were randomly assigned to home-based or gym-based climbing, with two climbers withdrawing because of illness within the first two weeks. The final participants were 26 home-based and 24 gym-based individuals. As all bar the fasting blood measurements were obtained in the field rather than the laboratory, a no-intervention, healthy weight and physically active control group (50 mins.wk^−1^ MVPA) with the same exclusion criteria was recruited to test for the robustness of effects on the outcome variables. Participants were asked not to change their diet or physical activity over the experimental period.

### 2.2. Measurements and Procedure

The measurements below were completed before and after the 8-week intervention period.

Body mass and height used the FitBit Aria Scale with the participants in light indoor clothing without shoes. Half a kilogram was deducted from the measured mass to account for clothing worn. Using Harbenden Skinfold Calipers, skinfold measurements were taken twice (triceps, chest, midaxillary, subscapular, suprailiac, abdominal, thigh), with the average of the measures used. Percentage of body fat was calculated using the 7-site skinfold equation [34].

After five minutes quiet sitting, heart rate and blood pressure (left radial) were measured twice at a 2-min interval with a wrist blood pressure monitor (Life source A&D Medical UB-328). The wrist was positioned at heart level with an armrest. The two readings were averaged.

For the fitness assessment, participants had a light breakfast one hour before the test (three whole-wheat crackers, 80 g cottage cheese, 10 mL honey, 250 mL fresh orange juice). To estimate cardiorespiratory fitness, the $\dot{V}$O_2_max proxy from the Multi-Stage Fitness Test (MSFT) which allows simultaneously testing of more than one participant was used [35]. The testing was carried out in groups of 3–4 participants. The relationship between MSFT score and estimated $\dot{V}$O_2_max has been found to be independent of sex and age [36]. The MSFT, also known as the ‘bleep test’, involves a 20 m shuttle run between two points. Each shuttle must be completed before a bleep is played over a loudspeaker. The time between each bleep progressively decreases, requiring participants to increase their pace to reach the point before the next bleep. When a participant was unable to complete a shuttle run before a bleep the test ended for that participant and the level and bleep recorded. For rating of perceived exertion (RPE), participants pointed with their finger at a large board displaying exertion levels on the Borg 6–20 scale [37]. The final RPE level at which a participant could not complete the shuttle run was recorded. Finger blood lactic acid was tested immediately after termination with a portable Accutrend Plus System capillary blood tester.

Assessment of leg power used the counter movement jump without arm swing. The time of flight was recorded with a Bosco Ergo jump System Mat. Participants began in an upright stance with arms on hips and when signalled, squatted to the 90° leg bend position and immediately jumped as high as possible. Participants were instructed to maintain their body posture and shape during the flight. Participants performed one practice trial and then three attempts with a small rest in between each (15 s). The highest jump was recorded. Height jumped in cm as an index of leg power was derived from the equations of Komi and Bosco (1978) [38].

Prior to blood sampling, participants ate a uniform evening meal (100 g grilled chicken breast, 50 g total low-fat yogurt, one cup chopped lettuce with cucumber, 8 g olive oil, one slice wholemeal bread) followed by a 12 h fast. All data were collected between 08:00 and 10:00. Plain vacutainer tubes were used for the collection of blood from an antecubital vein for measurement of total cholesterol, HDL, LDL and triglycerides whereas glucose measurements employed sodium fluoride tubes to prevent glycolysis. The blood samples were analysed with standard laboratory methods on a Cobas 400 plus Analyzer in a clinical laboratory subject to external quality control by the national external quality assessment scheme for clinical chemistry. LDL was quantified from direct measurement to minimise accumulated errors. The post-intervention blood sampling was completed a minimum of 60 h after the end of the intervention to avoid transient effects of exercise on serum lipids [39].

### 2.3. The Intervention Program

The study aimed to replicate the height of climb, 32.8 m, used by Boreham and co-workers [18,19]. For home-based climbers, programmes were individualised. For example, a 2-storey house with 20 steps of 17 cm riser height to reach the next floor required 9.6 ascents to climb 32.8 m, rounded to 10 floors of continuous climbing and descending for a single bout in this house. Participants chose their own pace to ascend and descend. The gym-based group climbed on a LEEKON Stair Machine (LK-7000) with a step height of 23 cm. Thus, a single bout required 142.6 steps rounded to 143 steps. Participants chose a starting pace between 50–60 steps.min^−1^ for each bout. If they subsequently felt comfortable climbing faster, they could increase it. Participants did not walk down on the machine as it was only designed for ascent.

For the 8-week intervention, participants started with two bouts of climbing, five days.wk^−1^ for the first two weeks. For each subsequent 2-week period, a further daily bout of climbing was added so that they progressed from two bouts.day^−1^ to five bouts.day^−1^ in weeks seven and eight of the intervention. Participants were reminded to increase their number of daily bouts at each transition by phone. Both home-based and gym-based participants were instructed to rest for a minimum of 10 min between each bout.

### 2.4. Statistical Analysis

Preliminary inspection of skew and kurtosis and outliers with boxplots revealed no major problems. Formal testing with the Wilks-Shapiro test (*p* < 0.05), however, revealed non-normal distributions for estimated $\dot{V}$O_2_max, RPE, triglycerides, weight and fat that could not be improved with a natural logarithmic transformation. While the *F* test in ANOVA is robust to violations of normality [40], non-parametric analyses for these variables are provided in Appendix A, as are the medians and interquartile ranges (IQR). These non-parametric analyses confirm the ANOVA results. As a result, ANOVA is retained in the main text because of its protection of the family-wise error rate for multiple comparisons and the more comprehensive analysis available with parametric statistics.

Analyses employed a two-stage process a priori. First, repeated measures analyses of variance with the within subject factor of pre vs. post compared the combined intervention groups with the controls. Second, repeated measures analyses of variance in the intervention groups alone with the between subject factor of location (gym vs. home) tested for the predicted effects pre vs. post and for any differences between the subgroups in the effects. Significant effects were followed up with paired *t*-tests with Bonferroni protected probabilities where required. Results are presented as mean ± *SE* and two-tailed probabilities reported for all statistical testing. Partial eta^2^ (*ɳ_p_*^2^) was employed as the measure of effect size.

## 3. Results

Table 1 below summarises the age, smoking status, BMI, estimated $\dot{V}$O_2_max and MVPA of the three recruited groups. The control group had lower BMI and participated in more MVPA than the stair groups (both *p* < 0.003). There were no differences between the home and gym-based climbers (all *p* > 0.36).

Comparisons between the controls and the two stair climbing groups at baseline, revealed that the climbers weighed more than the controls (*F*_2,57_ = 8.23, *p* = 0.001, *ɳ_p_*^2^ = 0.224) and had more body fat (*F*_2,57_ = 5.49, *p* = 0.007, *ɳ_p_*^2^ = 0.162). There were no other differences between the three groups at baseline.

### 3.1. Effects of Stair Climbing

All the tables below have a common structure. The left-hand side contains the means (*SE*) pre vs. post for control, gym-based and home-based stair climbing groups, with summaries of statistical testing on the right-hand side of the table. The first two columns of statistical testing summarise the main experimental tests; (1) stair climbing will be more healthful than making no change (Cnt:Exp x pre:post) and (2) stair climbing will reduce health risk for each variable (stair group pre:post). The final two columns summarise comparisons of gym-based and home-based climbing: (3) tests for overall differences between the randomized stair groups (climbing location) and (4) tests for different effects of continuous climbing in the gym and the alternating ascent and descent at home (location x pre:post). The results in these final two columns are reported in the next section.

As two-tailed probabilities are displayed throughout in the tables, *p* ≤ 0.10 represents the predicted effects. Presented below each *F*-ratio is partial eta squared (*ɳ_p_*^2^) as a measure of effect size. For clarification, 0.010 represents a small effect size, with 0.059 and 0.138 representing medium and large effect sizes, respectively. Table 2 summarises the results for variables related to estimated aerobic fitness.

As can be seen from the first summary column of statistical testing in the table, relative to controls, stair climbing improved both indices of aerobic fitness, namely $\dot{V}$O_2_max and lactate. The statistically significant improvements for these variables in the stair group shown in the next column were both associated with large effect sizes. The absence of differences in ratings of perceived exertion indicate that all groups were exercising at similar levels during the fitness test. In addition, there was an increase in the index of leg power, counter movement jump height, after the intervention in the stair group, though the effect did not differ statistically from the controls. Nonetheless, improvement in the stair group was also associated with a large effect size.

Table 3 summarises the results for serum lipids. As changes in total cholesterol confound potential increases in HDL with decreases in LDL, Non-HDL cholesterol was analysed. As can be seen from the table, stair climbing improved all measured serum lipids (HDL, LDL, Non-HDL cholesterol, triglycerides) relative to controls as predicted. Once again, the effect sizes were large.

Table 4 above summarises the results for metabolism-related variables. Relative to controls, stair climbing improved one index of body composition, namely weight. In addition, estimated body fat was reduced in the stair climbing group. Both these effects of stair climbing were large. There was no evidence of beneficial effects of stair climbing *overall* on fasting blood glucose (see Section 3.2).

Table 5 below summarises the results for resting cardiovascular variables. The medium effect size reduction in SBP confined to the stair group did not differ relative to the controls. For DBP, the differences between the controls and intervention group pre vs. post resulted from a paradoxical reduction in the control group (*p* = 0.01). There was no evidence of beneficial effects of stair climbing on DBP or resting heart rate.

### 3.2. Comparisons between Home-Based and Gym-Based Interventions

As noted earlier, the third column of statistical testing in the tables above summarises comparisons between the stair climbing groups overall, i.e., tests differences by randomization, whereas the final column tests for differential intervention effects for gym and home-based climbing. As can be seen, only one variable out of 13 provided any evidence of differential effects of intervention location (see Table 4). The effect of location for glucose resulted from the contrast between a significant reduction in the home-based group (*p* = 0.03) and a non-significant increase in the gym-based participants (*p* = 0.30) with Bonferroni adjusted probabilities.

Finally, for the test of randomization, location overall, the home-based group reported higher rates of perceived exertion at the end of each fitness test, suggesting that home-based climbers reached a higher level of exertion during testing overall. Nonetheless, similar improvements in aerobic fitness occurred in both groups (see Table 2). Apart from this difference, randomization produced similar groups for the variables.

Follow-up *t*-tests with Bonferroni adjustment for each location separately, confirmed significant improvements in aerobic fitness, serum lipids, and body composition in both stair climbing groups (all *p* < 0.03).

## 4. Discussion

### 4.1. Effects of Stair Climbing

The predicted improvements from stair climbing relative to controls on aerobic fitness ($\dot{V}$O_2_max, +6.4% ± 0.8; lactate, −19.1% ± 3.6), serum lipids (HDL, +17.3% ± 4.1; LDL, −5.4% ± 1.8; Non-HDL, −6.7% ± 1.8; triglycerides, −12.5% ± 2.3) and body composition (weight, −1.4% ± 0.3) provide strong statistical evidence of efficacy. In addition, improvements in climbers at the end of the intervention for percentage body fat (−7.8% ± 1.2) and leg power (+5.9% ± 1.8) provide weaker evidence. All these effect sizes for climbers were large. Climbing did not improve resting DBP, heart rate or fasting blood glucose overall. For resting cardiovascular variables, the pattern was mixed and not easily interpretable. Previous effects on blood pressure without controls [22] contrast with no effects in controlled studies [17,21].

Improvements in aerobic fitness from stair climbing are consistent with all previous tests [17,18,19,20,21,22,23]. Improved fitness when sedentary individuals increase physical activity is a likely outcome; this benefit can be achieved on stairs at home. The key question, however, was whether risk factors additional to fitness would change. Intervention improved all serum lipids. The target of five ascents.day^−1^ in the final two weeks was a similar volume of stair climbing to studies that affected some lipoproteins [18,19] but more than the three ascents.day^−1^ that had no effect [21]. Unlike all previous studies, stair climbing reduced the MetS risk factor of triglycerides (c.f. [19,22,23,30]). The standardised evening meal prior to an overnight fast here might have contributed to this effect. Nonetheless, the most likely explanation for comprehensive lipid effects was the greater statistical power available with 50 participants compared to the smaller sample sizes with equivalent heights of climb, *n* = 12 [18] and *n* = 8 [19].

For effects on body composition, previous controlled investigations reported no changes [17,18,19,21]. Here, the controlled comparisons confirm an uncontrolled report of reductions in weight and body fat [22]. The strong statistical evidence for improvements in weight, as well as weaker evidence for changes in body fat, are encouraging. Nonetheless, the study estimated fat from skinfolds rather than using the gold standard measure, Dual-Energy X-ray Absorptiometry (DEXA scanning). A recent study has confirmed that stair climbing and descent reduce fat mass with DEXA scanning for the first time [41]. Changes in weight, here, indicate that fat must have been lost. The exercise that clearly occurred could increase lean body mass, i.e., muscles (c.f. [17]), consistent with the increased leg power. Such a change would increase mass overall. Any weight loss would have to be primarily fat. Subsequent studies using DEXA scanning and food diaries during the intervention would be required to confirm these field estimates in the home.

In summary, the improvements in risk factors can be grouped into meaningful clusters. Changes in cholesterol and weight, additional to aerobic fitness, reduced CVD risk whereas improvements in body fat, triglycerides, and HDL cholesterol reduced MetS risk.

### 4.2. Effects of Intervention by Location

The comparisons between gym-based and home-based climbing tested for differential effects of repeated ascent and descent relative to continuous climbing [18,19,21]. The stair interventions were matched for height of the climb, i.e., the vigorous component of stair use [13]. Accumulation of subsequent descents in the home-based group would add further expenditure to their intervention. There were no generalised effects of this additional expenditure; equivalent improvements in aerobic fitness, serum lipids and body composition occurred in the stair groups. The solitary difference was metabolic.

Walking up and down stairs at home reduced fasting blood glucose (−3.9% ± 1.5). A similar protocol, walking up and down a single floor of stairs, improved post-prandial glycaemic control acutely in sedentary, middle-aged men [42] and Type 2 diabetics [43,44,45]. Two studies have piloted ascent/descent protocols as potential longer-term interventions for Type 2 diabetes. A six-week period of nine, 1-min bouts each week provided an insufficient dose of intervention for changes [46]. In contrast, a two-week, home-based intervention that entailed four, 3 min bouts after each meal, i.e., 36 min intervention each day, improved glycaemic control [30] though it did not significantly reduce fasting glucose. As with serum lipids, the greater sample size here for home-based climbing, 26, compared to samples of seven in the two pilot studies might explain some of the discrepancies. Nonetheless, a recent pilot of a four-floor ascending and descending protocol improved fasting glucose, as well as LDL, in a relatively small sample of randomly allocated climbers (*n* = 8; Suhana, Wallis, White and Eves, in preparation). Compared to continuous ascents, the descent component of stair use may be important. In formal comparisons, descent improved glycaemic control more than ascent [41,47]. The eccentric nature of exercise when descending may be an important bonus of a home-based intervention [41]. If confirmed in subsequent studies, the effects on glucose confined to the home-based group suggest an alternating ascent and descent protocol may offer an improvement on continuous climbing for MetS risk.

### 4.3. Alternative Stair Interventions for the Home

Relatively low volumes of increased stair climbing can improve cardiorespiratory fitness; additional accumulations of 24 [21] and 16 [22] floors.day^−1^ were effective. Two recent studies have employed a High Intensity Interval Training (HIIT) protocol that reduces the time required for exercise [48,49,50,51], adapting it as a time-efficient, stair climbing intervention [17,23]. These HIIT protocols had young participants climbing as ‘quickly as possible’ for a 3-floor climb [23] (p. 682) and, in the protocol most comparable to home-based climbing here, ‘vigorously but not all out’ when walking up and down one flight for a minute [17] (p. 300). The resultant speeds of ascent were 0.559 m.s^−1^ and 0.562 m.s^−1^ respectively. These fast climbs required warm-up and cool down periods and might not be considered comfortable by many participants [52]. While a scoping review suggested HIIT would be an acceptable intervention, 75% of the studies employed participants of 30 years or less [53]. If home-based stair climbing is to be a practical public health intervention, speed of climbing may be critical to translation, particularly for older participants. Cardiorespiratory fitness was increased at brisk but comfortable speeds, 0.243 m.s^−1^ in the initial study [18], and 0.247 m.s^−1^ for a young sample [19] and 0.205 m.s^−1^ for an older sample [21] in subsequent studies. Here, participants selected a speed for climbing that suited them.

Stair climbing is a vigorous activity because it requires participants to raise *all* their body mass against gravity. Energy expenditure is proportional to mass raised and speed of climbing is a relatively small contributor to metabolic cost. Climbing slowly, e.g., 60 steps.min^−1^ equivalent to 0.203 m.s^−1^, cost 8.6–8.7 METs ([54], Eves and White unpublished) whereas climbing briskly at almost twice that rate, 110 steps.min^−1^ or 0.275 m.s^−1^, cost 9.6 METs [13]. In the lowest volume controlled study at a comfortable speed, 0.205 m.s^−1^, participants still only climbed for about six minutes, five days a week [21]. On average, this was accumulated in three, 2-min bouts. The subsequent stair descent would be quicker but at a lower, moderate intensity [13,54]. No warm-up or cool-down periods were included. In total, an average time commitment of 11 min each day was required of participants [21]. Stair climbing is a vigorous physical activity, even at slow speeds, that can increase cardiorespiratory fitness with a lower time commitment than conventional physical activities [18,19,21]. The haste of recent HIIT protocols is unnecessary for fitness, irrespective of any beneficial changes in muscle metabolism of elevated speed [48].

### 4.4. Limitations and Future Directions

Fitness was improved but the magnitude of $\dot{V}$O_2_max change was estimated from shuttle running rather than direct measurement of respiration and heart rate. This field estimate is to exhaustion, (c.f. [17,19,23]), unlike submaximal tests [18,20,21,22], and testing took place at the same time of day with a fixed prior food intake, (c.f. [17,22]). Nonetheless, repeated changes in direction when shuttle running disproportionately affect those who have more mass to decelerate and accelerate when they change direction. Participants can increase the estimate by improving the underlying motor skill with practice, requiring comparison against a control group as was performed. Had $\dot{V}$O_2_max been estimated from cycle ergometry, the absolute magnitude of improvement might have differed. Subsequent studies should test the magnitude of the effects on aerobic fitness using cycle ergometry to exhaustion.

The estimates of fitness and skinfolds measures for body fat, were not ‘gold standard’ measures. Rather they were simple field estimates that could be used by health and fitness advisers. Field measures allow public health to test the efficacy of their interventions, (c.f. [55]). While a randomised group was not employed, the controls tested for the robustness of the field procedures; neither estimate of improvements in body fat or leg power differed from control in the parametric analyses. Subsequent studies should confirm these effects with random allocation and greater precision of measurement. In particular, DEXA scanning and food diaries during the intervention in subsequent studies would be more informative about body fat and weight loss.

One omission was that participants did not keep logs of their climbing bouts unlike the studies by Boreham and co-workers [18,19,21]. Logging would be helpful for intervention success because it provides the self-monitoring of behaviour that facilitates change [56] and should be included in any follow-up. Clearly, climbing took place but intervention adherence was unknown. Had there been major differences in results between the home and gym-based groups, differing adherence would have provided an alternative explanation for any differences. Finally, the sample was exclusively female. Most previous research on the physiological benefits of stair use interventions has tested only female participants [17,18,19,41,47]. Improvements in fitness in samples including males have been reported [20,21,23] and increases in fitness seem likely in sedentary individuals who increase their exercise levels, irrespective of sex. Nonetheless, only a single study assessed glycaemic control in males [30]. Subsequent studies should attempt replication of the results reported here in males to allow generalisation.

### 4.5. Strengths

Home and gym-based interventions had similar effects. For the solitary differential effect, i.e., fasting glucose, the outcome favoured the alternating ascent and descent protocol of home-based climbing. Put another way, home-based climbing was at least as effective as the gym-based alternative for 13 different variables, seven of which showed the predicted improvements relative to control. The benefits of physical activity are well documented. What was novel here was that individuals comfortably walking up and down stairs at home reduced not only CVD risks, but also the MetS risk factors of triglycerides and fasting glucose, as well as increasing HDL. These clusters of improvement occurred in a relatively low-fit group for their age ($\dot{V}$O_2_max = 25.3 mL.min^−1^.kg^−1^ ± 0.6). Greater public health outcomes occur for improvements in the less fit [27,57].

There are few logistical barriers to stair climbing at home, unlike gym-based exercise. It is a simple lifestyle activity accumulated in short bouts. Based on the times in Teh and Aziz’s (2002) study, and unpublished data, the combined ascent and descent would require less than four minutes, about the length of a commercial break in a television programme. Stair climbing requires no equipment, specialised clothing or sporting ability. It is non-competitive, unlike sport, avoiding a potential barrier for the less accomplished. All buildings bar bungalows have a staircase.

Most of the population can climb stairs because they already do so as part of daily life. As a result, most of the population believe they can climb stairs. Belief that one can perform a behaviour, called self-efficacy, is a major determinant of actual performance for behaviour in general [58] and for physical activity [59,60,61,62]. Self-efficacy develops primarily from experience of the behaviour; even a single session of a challenging physical activity such as walking quickly for the elderly, increases self-efficacy for that physical activity after the first attempt [59]. The progressive increase in daily bouts here, at a speed selected by participants, would facilitate development of efficacy for climbing. Self-efficacy predicts future physical activity [59,61] and an incremental experience of success is beneficial for its development [58]. As most of the population already climb stairs, low self-efficacy is unlikely to be a barrier to participation as it can be for formal sport or exercise. Further, if physical activity is performed at home, participants avoid most of the logistical barriers to arranging their physical activity such as travel to the venue and exercise partners that might impair self-efficacy [61].

As noted in the introduction, CVD required 8% and 17% of health care expenditure in the EU and US respectively [2,3], with diabetes costing 10% and 14% in the EU and US [6,7]. Here, home-based stair use produced clusters of reduced risk for cardiovascular disease and MetS. Clearly, increased stair climbing reduces risk of disease and all-cause mortality as indicated by the Harvard Alumni studies [14,15,16]. For walking up and down stairs at home, the potential economic costs to health services are low, as they are for participants, suggesting home-based stair climbing offers a low-cost approach for disease risk reduction to public health.

### 4.6. Conclusions

In this study, the expected improvements in fitness from climbing were accompanied by improvements in all serum lipids and body composition, as well as reductions in fasting glucose for home-based participants. Home-based climbing was at least as effective as an equivalent gym-based protocol for all the study variables. The clusters of improvements here were achieved at an in-study cost of four individual phone calls to participants over the eight-week intervention. Walking up and down stairs at home reduces health risk at low cost for the individual.

## Figures and Tables

**Table 1 ijerph-18-00603-t001:** Characteristics of the three recruited groups.

Variable	Control (*SE*) (*n* = 10)	Gym (*SE*) (*n* = 24)	Home (*SE*) (*n* = 26)
Age (years)	32.00 (2.75)	31.58 (1.41)	31.76 (1.33)
Smoking n (%)	5 (50)	8 (33)	12 (46)
BMI	20.59 (0.59)	26.33 (1.18)	27.80 (1.11)
$\dot{V}$O_2_max (ml.min^−1^.kg^−1^)	25.50 (1.32)	25.67 (0.85)	24.98 (0.86)
Weekly MVPA ^a^ (min)	52.00 (28.12)	9.38 (2.63)	10.96 (2.04)

^a^ MVPA = moderate to vigorous physical activity.

**Table 2 ijerph-18-00603-t002:** Means (*SE*) pre and post for the control, gym-based and home-based groups for fitness variables, with a summary of statistical testing.

	Control (*n* = 10)		Gym-Based (*n* = 24)		Home-Based (*n* = 26)		Cnt:Exp ^a^ x Pre:Post *F*_1,58_ Effect Size	Stair Group Pre:Post *F*_1,48_ Effect Size	Climbing Location *F*_1,48_ Effect Size	Location x Pre:Post *F*_1,48_ Effect Size
Variable	Pre (*SE*)	Post (*SE*)	Pre (*SE*)	Post (*SE*)	Pre (*SE*)	Post (*SE*)				
$\dot{V}$O_2_max ^d^ (ml.min^−1^.kg^−1^)	25.50 (1.32)	26.03 (1.52)	25.68 (0.85)	27.28 (0.99)	24.99 (0.86)	26.64 (0.98)	**6.01 *^bc^** ***ɳ*_p_^2^ = 0.10**	**64.50 ***** ***ɳ*_p_^2^ = 0.58**	0.34 *ɳ*_p_^2^ = 0.01	05 *ɳ*_p_^2^ = 0.00
Lactate (mmol.L^−^^1^)	10.52 (0.71)	9.76 (0.90)	11.20 (0.70)	8.90 (0.69)	12.58 (1.04)	9.70 (0.84)	2.17 *ɳ*_p_^2^ = 0.04	**25.45 ***** ***ɳ*_p_^2^ = 0.35**	0.33 *ɳ*_p_^2^ = 0.01	0.03 *ɳ*_p_^2^ = 0.00
Rating of perceived exertion ^d^	16.80 (0.80)	17.00 (0.82)	15.63 (0.82)	15.92 (0.74)	17.58 (0.54)	17.81 (0.32)	0.01 *ɳ*_p_^2^ = 0.00	0.55 *ɳ*_p_^2^ = 0.01	**5.64 *** ***ɳ*_p_^2^ = 0.11**	0.01 *ɳ*_p_^2^ = 0.00
Counter movement jump height (cm)	20.73 (1.52)	20.40 (1.60)	21.32 (1.42)	22.70 (1.71)	18.21 (1.12)	19.05 (1.13)	2.01 *ɳ*_p_^2^ = 0.03	**8.37 **** ***ɳ*_p_^2^ = 0.15**	3.45 *ɳ*_p_^2^ = 0.07	0.01 *ɳ*_p_^2^ = 0.00

^a^: cnt = control; exp = experimental. ^b^: * *p* ≤ 0.05, ** *p* ≤ 0.01, *** *p* ≤ 0.001. ^c^: Significant effects with two-tailed probabilities are presented in bold. As two-tailed probabilities are employed throughout. ^d^: Non-parametric analyses are available in Appendix A.

**Table 3 ijerph-18-00603-t003:** Means (*SE*) pre and post for the control, gym-based and home-based groups for serum lipids, with a summary of statistical testing.

*9*	Control (*n* = 10)		Gym-Based (*n* = 24)		Home-Based (*n* = 26)		Cnt:Exp ^a^ x Pre:Post *F*_1,58_ Effect Size	Stair Group Pre:Post *F*_1,48_ Effect Size	Climbing Location *F*_1,48_ Effect Size	Location X Pre:Post *F*_1,48_ Effect Size
Variable	Pre (*SE*)	Post (*SE*)	Pre (*SE*)	Post (*SE*)	Pre (*SE*)	Post (*SE*)				
HDL cholesterol (mmol.L^−1^)	1.58 (0.06)	1.64 (0.06)	1.41 (0.76)	1.51 (0.07)	1.39 (0.08)	1.54 (0.08)	3.56 †^bc^ *ɳ*_p_^2^ = 0.06	**65.05 ***** ***ɳ*_p_^2^ = 0.58**	0.00 *ɳ*_p_^2^ = 0.00	3.39 *ɳ*_p_^2^ = 0.06
LDL cholesterol (mmol.L^−1^)	2.74 (0.26)	2.88 (0.22)	3.31 (0.20)	3.10 (0.21)	3.48 (0.24)	3.28 (0.22)	**7.21 **** ***ɳ*_p_^2^ = 0.11**	**13.80 ***** ***ɳ*_p_^2^ = 0.22**	0.33 *ɳ*_p_^2^ = 0.01	0.01 *ɳ*_p_^2^ = 0.00
Non-HDL cholesterol (mmol.L^−1^)	2.93 (0.24)	2.95 (0.24)	3.32 (0.20)	3.09 (0.21)	3.63 (0.26)	3.38 (0.24)	3.70 † *ɳ*_p_^2^ = 0.06	**16.88 ***** ***ɳ*_p_^2^ = 0.26**	0.89 *ɳ*_p_^2^ = 0.02	0.03 *ɳ*_p_^2^ = 0.00
Triglycerides ^d^ (mmol.L^−1^)	0.93 (0.09)	0.97 (0.15)	1.30 (0.14)	1.14 (0.11)	1.43 (0.10)	1.16 (0.08)	**10.87 **** ***ɳ*_p_^2^ = 0.16**	**50.54 ***** ***ɳ*_p_^2^ = 0.51**	0.22 *ɳ*_p_^2^ = 0.00	3.50 *ɳ*_p_^2^ = 0.07

^a^: cnt = control; exp = experimental. ^b^: † *p* ≤ 0.10, ** *p* ≤ 0.01, *** *p* ≤ 0.001. ^c^: Significant effects with two-tailed probabilities are presented in bold. As two-tailed probabilities are employed throughout, *p* ≤ 0.10 represents the predicted effects (†). ^d^: Non-parametric analyses are available in Appendix A.

**Table 4 ijerph-18-00603-t004:** Means (*SE*) pre and post for the control, gym-based and home-based groups for metabolism-related variables, with a summary of statistical testing.

*9*	Control (*n* = 10)		Gym-Based (*n* = 24)		Home-Based (*n* = 26)		Cnt:Exp ^a^ x Pre:Post *F*_1,58_ Effect Size	Stair Group Pre:Post *F*_1,48_ Effect Size	Climbing Location *F*_1,48_ Effect Size	Location x Pre:Post *F*_1,48_ Effect Size
Variable	Pre (*SE*)	Post (*SE*)	Pre (*SE*)	Post (*SE*)	Pre (*SE*)	Post (*SE*)				
Weight (kg) ^d^	52.87 (1.45)	52.91 (1.49)	69.39 (3.03)	68.26 (3.03)	74.35 (3.12)	73.49 (3.14)	**5.41 *^bc^** ***ɳ*_p_^2^ = 0.09**	**26.10 ***** ***ɳ*_p_^2^ = 0.35**	1.37 *ɳ*_p_^2^ = 0.03	0.51 *ɳ*_p_^2^ = 0.01
Body fat (%) ^d^	22.52 (1.15)	21.79 (1.14)	27.21 (1.18)	25.20 (1.28)	29.79 (1.31)	27.46 (1.32)	2.58 *ɳ*_p_^2^ = 0.04	**30.77 ***** ***ɳ*_p_^2^ = 0.39**	1.97 *ɳ*_p_^2^ = 0.04	0.18 *ɳ*_p_^2^ = 0.00
Fasting Glucose (mmol.L^−1^)	5.00 (0.12)	5.03 (0.07)	5.00 (0.09)	5.09 (0.08)	5.06 (0.06)	4.86 (0.09)	0.52 *ɳ*_p_^2^ = 0.01	1.18 *ɳ*_p_^2^ = 0.02	0.81 *ɳ*_p_^2^ = 0.02	**8.31 **** ***ɳ*_p_^2^ = 0.15**

^a^: cnt = control; exp = experimental. ^b^: * *p* ≤ 0.05, ** *p* ≤ 0.01, *** *p* ≤ 0.001. ^c^: Significant effects with two-tailed probabilities are presented in bold. As two-tailed probabilities are employed throughout. ^d^: Non-parametric analyses are available in Appendix A.

**Table 5 ijerph-18-00603-t005:** Means (*SE*) pre and post for the control, gym-based and home-based groups for resting cardiovascular variables, with a summary of statistical testing.

	Control (*n* = 10)		Gym-Based (*n* = 24)		Home-Based (*n* = 26)		Cnt:Exp ^a^ x Pre:Post *F*_1,58_ Effect Size	Stair Group Pre:Post *F*_1,48_ Effect Size	Climbing Location *F*_1,48_ Effect Size	Location x Pre:Post *F*_1,48_ Effect Size
Variable	Pre (*SE*)	Post (*SE*)	Pre (*SE*)	Post (*SE*)	Pre (*SE*)	Post (*SE*)				
Systolic blood pressure (mmHg)	115.60 (2.16)	114.00 (2.42)	121.71 (2.29)	118.88 (2.64)	120.27 (2.06)	118.70 (2.04)	0.05 *ɳ*_p_^2^ = 0.00	**4.18 *^bc^** ***ɳ*_p_^2^ = 0.08**	0.07 *ɳ*_p_^2^ = 0.00	0.33 *ɳ*_p_^2^ = 0.01
Diastolic blood pressure (mmHg)	79.00 (1.85)	72.60 (2.48)	77.04 (1.21)	76.42 (1.53)	75.50 (1.49)	74.50 (1.89)	**8.15 **** ***ɳ*_p_^2^ = 0.12**	1.25 *ɳ*_p_^2^ = 0.03	0.70 *ɳ*_p_^2^ = 0.01	0.06 *ɳ*_p_^2^ = 0.00
Resting heart rate (bpm)	82.90 (3.58)	81.60 (2.33)	86.00 (1.57)	82.54 (1.24)	84.19 (2.74)	83.96 (2.20)	0.02 *ɳ*_p_^2^ = 0.00	1.60 *ɳ*_p_^2^ = 0.03	0.01 *ɳ*_p_^2^ = 0.00	1.22 *ɳ*_p_^2^ = 0.02

^a^: cnt = control; exp = experimental. ^b^: * *p* ≤ 0.05, ** *p* ≤ 0.01. ^c^: Significant effects with two-tailed probabilities are presented in bold. As two-tailed probabilities are employed throughout.

## Data Availability

The full data set is available as Appendix A.

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
