# Peer review of "Home-Based Stair Climbing as an Intervention for Disease Risk in Adult Females; A Controlled Study"

_ijerph, 2021, doi:10.3390/ijerph18020603_

Round 1

Reviewer 1 Report

Due to fact of lack of novelty of this data, I strongly indicate the rejection of manuscript.

Reviewer 2 Report

  1. Climbing stairs for cardiovascular disease is a great exercise. In this paper, we compared the effects of continuous gym-based stair climbing and climbing stairs at home on disease risk factors of aerobic exercise, serum lipids, body composition, fasting blood glucose and resting cardiovascular variables. 
  2. Of course, you may find that going up and down stairs at home will reduce your costs.
  3. These results are weak as the contribution of the paper
  4. I need a figure to show the overall flow of this paper.
  5. It is necessary to write the conclusions in more detail and describe future work as well.

Reviewer 3 Report

This study compared the effects of aerobic exercise, serum lipids, body composition, fasting blood glucose, and resting on the effects of gym-based stair climbing and climbing stairs at home for risk factors for cardiovascular disease in adult women. My previous review of this manuscript (ID ijerph-1017822) pointed out several issues. These included the test for normality on the sample and other issues in the statistical analysis, along with improvement needed in presenting results and the Conclusions section. The revised manuscript (ID ijerph-1062707) addressed all these issues adequately. There is only one final recommendation for the title: to indicate that the participants were female adults, so it reads: “Home-based stair climbing as an intervention for disease risk in female adults: a controlled study”.

Round 2

Reviewer 1 Report

I agree with comments and responses provided by authors. These data shown a large effect size which are very novelty and important for readers and this journal. 

This manuscript is a resubmission of an earlier submission. The following is a list of the peer review reports and author responses from that submission.

Round 1

Reviewer 1 Report

This manuscript evaluated the effect of an 8-week exercise program on fitness variables, serum lipids, and metabolism-related variables. The study was performed on female adults, one group performed stair climbing at home, another performed stair climbing at the gym, and one control group as reference. The work is interesting and original, and the manuscript is well-written. There are only a few issues as described below.

Major issues

  • Statistical analysis. Which test was employed to verify normal distribution in each variable? In variables where normal distribution cannot be assumed, the descriptive tests and the inferential tests must be non-parametric.
  • Page 10, lines 205 to 251. The follow-up t-tests for each location separately are not appropriate, as the study design has three groups that were compared between them with the ANOVA analysis. Instead, post-hoc analysis should be performed to estimate the p-values adjusted by the Bonferroni method or other ones that may be appropriate.
  • The study was performed only on female adults and this is not justified in the Introduction section, and it is not discussed adequately (there is only a phrase mentioning this as a study limitation). The inclusion of only female participants in the study is so important, that it should be mentioned in the title.
  • Power calculations, page 96. Please specify the type of statistical test and the measure of improvement used to calculate the sample size with the program GPower.
  • Section 2.1. Participants. Please define the selection criteria of the healthy weight control group.
  • Conclusions section. All statements in the conclusions should be based only on the results of the present study, all the phrases referring to previous studies are only appropriate in the Discussion section. The last two statements, for instance, are not conclusions of the present study.

Minor issues

  • Line 100. Please provide the protocol number approved by the ethics subcommittee.
  • Line 152. Replace “bloods” with “blood samples”.

Reviewer 2 Report

This study compared the effects of aerobic exercise, serum lipids, body composition, fasting blood glucose, and resting on the effects of gym-based stair climbing and climbing stairs at home for risk factors for cardiovascular disease. 

  1. Cardiovascular disease and climbing stairs in this paper are a topic that has been used too much and a newer approach is needed. For example, the effect according to age, the effect of the patient and the normal person...
  2. In the experiment of the paper, the number of samples is too small. Analyzing with more sufficient data can yield more accurate results.
    You also need a test for normality on the sample.
  3. It is not specified how much each risk factor improved compared to the control data. Analyzing using logistic regression can show how effective it is.
  4. In the table 2, did F1,58 ANOVA analysis?  You need to be clear about how you analyze it.

Reviewer 3 Report

I read with great interest. This study compared effects of walking up and down stairs at home with continuous, gym-based stair climbing on the disease risk factors of aerobic fitness, serum lipids, body composition, fasting blood glucose and resting cardiovascular variables. However, severa issues are addressed:

-What is the sample sie? No clear evidence was provided in the methods. This study had a small sample.

-What are the novelty of these data? Data are known by literature scientific.

-I suggest to reduce the introduction.

-The authors must to add the data about diet and physical activity over the experimental time.

-The skinfold measurements are very variable between evaluations. This is a limitation (intra-inter variations)

-What are complete method of the blood lactic acid quantification?

-RPE is a subject measurement. It is a limitation?

In summary, due to fact of lack of novelty of this data, are strongly indicate the rejection of manuscript.